# A large scale test of the gaming-enhancement hypothesis

Andrew K. Przybylski[1] and John C. Wang[2]

[1] Oxford Internet Institute, University of Oxford, Oxford, United Kingdom
[2] Nanyang Technological University, Singapore

## ABSTRACT

A growing research literature suggests that regular electronic game play and game-based training programs may confer practically significant benefits to cognitive functioning. Most evidence supporting this idea, the *gaming-enhancement hypothesis*, has been collected in small-scale studies of university students and older adults. This research investigated the hypothesis in a general way with a large sample of 1,847 school-aged children. Our aim was to examine the relations between young people's gaming experiences and an objective test of reasoning performance. Using a Bayesian hypothesis testing approach, evidence for the gaming-enhancement and null hypotheses were compared. Results provided no substantive evidence supporting the idea that having preference for or regularly playing commercially available games was positively associated with reasoning ability. Evidence ranged from equivocal to very strong in support for the null hypothesis over what was predicted. The discussion focuses on the value of Bayesian hypothesis testing for investigating electronic gaming effects, the importance of open science practices, and pre-registered designs to improve the quality of future work.

# INTRODUCTION

## Background

Electronic games are now a ubiquitous form of entertainment and it is popularly believed that the time spent playing games might have positive benefits that extend outside of gaming contexts (*Lenhart et al., 2008*; *McGonigal, 2012*). This general idea, the *gaming-enhancement hypothesis*, posits that electronic gaming contexts influence a range of perceptual and cognitive abilities because they present complex, dynamic, and demanding challenges. A recent representative study of British adults and young people suggests this idea is widely held; Nine in ten think brain-training games provide an effective way to improve focus, memory, and concentration, and as many as two thirds have tried using these games to improve their own cognitive abilities (*Clemence et al., 2013*). This view is increasingly profitable and controversial (*Nuechterlein et al., 2016*). Given the intense public and private interest and investment in games as a way of improving cognition and reasoning, is noteworthy that the scientific literature investigating the gaming-enhancement hypothesis, though promising, is still at an early stage.

Corresponding author
Andrew K. Przybylski,
andy.przybylski@oii.ox.ac.uk

A growing body of research suggests that common varieties of electronic gaming experience might enhance general cognitive skills and abilities. These studies show that those who opt to, or are assigned to, play a range of commercially available games show measurable differences in terms of their visual and spatial abilities (*Quaiser-Pohl, Geiser & Lehmann, 2006*; *Green & Bavelier, 2006*), executive functioning, information processing (*Maillot, Perrot & Hartley, 2012*), and performance at specialist skills such as laparoscopic surgery (*Rosser et al., 2007*). In particular, action games (*Green & Bavelier, 2006*), strategy games (*Basak et al., 2008*), and multiplayer online games (*Whitlock, McLaughlin & Allaire, 2012*), have been identified as having enhancing effects because they provide complex multi-tasking environments that require players to integrate a range of sensory inputs. Researchers argue these environments lead players to implement adaptive strategies to meet the varied demands of these virtual contexts (*Bavelier et al., 2012*).

There is good reason to think that predispositions to engage specific kinds of games may relate to cognitive performance and reasoning more broadly. For example, in studying skill acquisition among strategy game players, researchers have reported evidence that differences in brain volume are correlated with speed and performance in gaming contexts (*Basak et al., 2011*). Exploration and learning in online gaming contexts closely mirror their offline analogues. Those with pre-existing strengths tend to thrive at online gaming challenges initially, but those with low levels of starting ability quickly close the performance gap (*Stafford & Dewar, 2013*). It is possible that such inclinations guide players to games that suit them. Findings from experimental and quasi-experimental studies suggest both preference and experience matter in terms of small to moderate effects across a wide range of cognitive performance indicators (*Powers et al., 2013*).

Unfortunately, many of these studies have pronounced shortcomings that temper the broad promise of the gaming-enhancement hypothesis (*Van Ravenzwaaij et al., 2014*). For example, most of the evidence supporting this view has been derived from small-scale surveys, intervention studies in university settings, or from samples of older adults. A recent review of this literature indicates a small yet consistent link between gaming and general reasoning ability (Cohen's $d = 0.24$), but the average sample size of studies examining the gaming-enhancement hypothesis is only 32 participants (*Powers et al., 2013*). Perhaps as a consequence of such small samples, the effects sizes reportedly linking games to cognitive abilities vary widely as a function of the methods used and the research groups investigating them. For example, studies published in top-tier journals, while typically using very small samples, report substantially larger effects compared to the rest of the literature (e.g., Cohen's $d = 0.85$; *Powers et al., 2013*). Moreover, despite the fact that games are played by the overwhelming majority of young people (*Lenhart et al., 2015*), fewer than one in ten studies of the gaming enhancement study have included participants under the age of 18. Childhood and adolescence see profound development in cognitive abilities, and though negative effects of games are fiercely debated (*Mills, 2014*; *Bell, Bishop & Przybylski, 2015*), there is very little evidence concerning their possible positive effects in this cohort. Further, nearly all studies examining gaming effects do so by studying individual games or game types in isolation. So although there

is reason to think that action (*Green & Bavelier, 2006*), strategy (*Basak et al., 2011*), and online game play (*Whitlock, McLaughlin & Allaire, 2012*) could have positive effects it is not possible to know if attitudes, preferences, or engagement with games in general, or specific subtypes in particular, are driving the effects reported. Finally, there are a number of larger-scale intervention studies that show evidence that does not support or directly contradicts the gaming-enhancement hypothesis (*Chisholm et al., 2010*; *Kennedy et al., 2011*). Given the nature of the existing evidence, research that systematically addresses these gaps in our knowledge is needed.

## Present research

The aim of the present research was to evaluate the extent to which everyday electronic game engagement by young people relates to their general reasoning abilities. In particular, we were interested to test how personal preferences for specific types of games related to reasoning ability. In line with previous research, we hypothesized that those who gravitate towards action, online, and strategy games would show higher levels of cognitive performance on a test of deductive reasoning ability (*Basak et al., 2011*). Our second goal was to test whether regular active engagement with action, online, and strategy games was linked to cognitive ability. In line with highly cited work in the area, we hypothesized that those playing these challenging games for more than five hours each week (*Green & Bavelier, 2006*) would show higher levels of reasoning ability.

## METHOD

### Data source

The present study utilized data collected in the first year of the Effects of Digital Gaming on Children and Teenagers in Singapore (EDGCTS) project. This dataset has been used in numerous previous publications focusing on the effects of gaming on motivation, dysregulated behavior, and interpersonal aggression in young people (*Wan & Chiou, 2006*; *Gentile et al., 2009*; *Wang, Liu & Khoo, 2009*; *Chen et al., 2009*; *Choo et al., 2010*; *Gentile et al., 2011*; *Wang et al., 2011*; *Li, Liau & Khoo, 2011*; *Chng et al., 2014*; *Prot et al., 2014*; *Gentile et al., 2014*; *Busching et al., 2015*; *Eichenbaum & Kattner, 2015*; *Chng et al., 2015*; *Liau et al., 2015b*; *Choo et al., 2015*; *Liau et al., 2015a*). A subsample of data from the EDGCTS project was used for the present study because it included self-reports of game play and an objective test of participants' reasoning abilities. Neither of these variables has been the focus of previously published studies, and a complete list of publications based on the EDGCTS dataset can be found on the Open Science Framework (osf.io/je786).

### Participants, ethics, and data

Ethical clearance for data collection was granted through the Institutional Review Board of Nanyang Technological University in Singapore. Because of the combined length of the EDGCTS testing protocol, data collection was conducted over a four-day period to reduce participant burden and minimize sequence effects. The research was deemed low risk and consent was obtained from parents through liaison teachers. Participants were informed that involvement in the project was voluntary and that they could withdraw at

any time. Ethical review for this secondary data analysis was conducted by the research ethics committee at the University of Oxford (SSH/OII/C1A-16-063).

In the first wave of the EDGCTS, quantitative data were collected from a total of 3,034 respondents. Gender information was present for 3,012 participants (99.3% of all cases), age data were available for 2,813 participants (92.7%), 2,135 participants provided the name of at least one electronic game they played (70.4%), and a total of 2,647 participants completed the reasoning test (87.2%). In sum, a total of 1,847 participants (60.9% of all cases), provided valid data and were included in subsequent analyses. These 1,847 participants ranged in age from 8 to 16 years ($M = 10.97$, $SD = 1.99$); 430 of these identified as female and 1,417 identified as male. The self-report materials, datasets, source code, and analysis code used in this study are available from the Open Science Framework (osf.io/je786).

## Outcome variable
### Reasoning ability
Participants' deductive reasoning ability was assessed using the 60-item Raven's Standard Progressive Matrices Plus (RPM) task (*Raven, Raven & Court, 2003*). The RPM, a widely used non-verbal test of reasoning ability, measures deductive intelligence by prompting takers to identify the key missing visual element that completes patterns shown in a series of increasingly complex $2 \times 2$, $2 \times 3$, $3 \times 3$, $4 \times 4$, and $6 \times 6$, matrices. This assessment was used because it has been well validated across a range of demographic and cross-cultural cohorts (*Raven, 2000*). Because our participants were school-aged children, they completed the version of this multiple-choice test designed for group administration in educational settings for students between the ages of 8 and 16 years. Participants got a median of 29 of 60 matrices ($SD = 4.44$) correct and age-adjusted reasoning scores were created for each participant in line with best-practices (*Savage-McGlynn, 2012*). To this end, participants were segregated by age and their raw performance scores were transformed into $z$-scores such that their performance was standardized with respect to other children their age.

## Explanatory variables
Participants' electronic gaming was assessed through a series of questions asking about the games they frequently played. Participants were requested to provide the names of up to three games they played as well as an estimate of the amount of time they spend playing each. A total of 446 (24.1%) participants named a single game, 485 (26.3%) participants named two games, and 916 (49.6%) named three games. The titles of named games were content coded to mark if they belonged to one of the three game categories of interest: action games (e.g., Call of Duty, Halo), multiplayer online games (e.g., Maple Story, World of Warcraft), or strategy games (e.g., SimCity, StarCraft).

### Game preference
Preference scores were created for each participant using the game names provided through self-report. If one or more of a participant's named games belonged to the action, online, or strategy types, they were marked as expressing a preference for this kind of
**Table 1  Participant game preferences and engagement.**

| | Game preference | | | Regular play | | |
|---|---|---|---|---|---|---|
| | **Action** | **Strategy** | **Online** | **Action** | **Strategy** | **Online** |
| Males | 21.7% | 25.1% | 45.1% | 13.0% | 17.2% | 33.7% |
| Females | 2.3% | 11.4% | 52.8% | 1.9% | 7.4% | 41.8% |
| Total | 17.2% | 21.9% | 47.2% | 10.4% | 14.9% | 34.1% |

game (coded 1); if not, they were counted as not having a preference for this game type (coded 0). This meant three preference scores, one for each game type, were computed for each participant.

### Regular play

Data from game preference scores and participants' self-reported play behavior were used to determine if participants were regular players of specific game types. Scores were created for participants by combing information about the kinds of games they expressed preferences for and their self-reported amounts of weekly play. Amounts of weekly play were computed summing participant estimates of weekday engagement, multiplied by five, and estimates of weekend day engagement multiplied by two. Codes for game types were then used to create one game engagement score for each type of game. In line with the approach prescribed in previous research (*Green & Bavelier, 2006*), participants were considered regular active players of a game type if they invested five or more hours in a given game type in a week (coded 1), and were coded 0 if they did not spend any time playing this game type in a typical week. Table 1 presents summary statistics for participant game preferences and proportions of active players of each game type.

## RESULTS

### Preliminary analyses

Zero-order bivariate correlations between observed variables are presented in Table 2. Because 36 correlations were conducted, we adjusted our $p$ value threshold for rejecting the null hypothesis from 0.05 to 0.0014 (*Holm, 1979*). Analyses indicated that older participants were more likely to report higher levels of engagement with games as compared to younger ones ($r$s = .130–.278). Similarly, female participants were less likely to engage action and strategy games ($r$s = .128–.217). Gender was not significantly related to online game play nor was it related to age-adjusted reasoning ability (all $p$s > 0.0014). Because gender was not associated with our target outcome, which was centered on age, neither age nor gender were considered as covariates in hypothesis testing.

### Game preference and reasoning ability

In line with meta-analytic evidence a series of one-sided Bayesian independent samples $t$-tests were used to quantify evidence for the game-enhancement hypothesis (*Rouder et al., 2012*; *Powers et al., 2013*; *Morey, Romeijn & Rouder, 2016*), that specified preference for action, online, and strategy games would be related to better deductive reasoning ability. Table 3 presents a summary of these results and observed means using a Cauchy prior

**Table 2  Observed zero-order correlations.**

| | | 2. | 3. | 4. | 5. | 6. | 7. | 8. | 9. |
|---|---|---|---|---|---|---|---|---|---|
| 1. Age | Pearson's r | −0.070 | 0.132[*] | 0.175[*] | 0.130[*] | 0.125[*] | 0.215[*] | 0.198[*] | −0.000 |
| 2. Female | Pearson's r | — | −0.217[*] | −0.140[*] | 0.062 | −0.167[*] | −0.126[*] | 0.040 | −0.029 |
| 3. Action game preference | Pearson's r | | — | 0.078[*] | −0.123[*] | 0.943[*] | 0.067 | −0.111[*] | −0.053 |
| 4. Strategy game preference | Pearson's r | | | — | −0.061 | 0.052 | 0.959[*] | −0.060 | 0.041 |
| 5. Online game preference | Pearson's r | | | | — | −0.093[*] | −0.046 | 0.946[*] | 0.051 |
| 6. Regular action game play | Pearson's r | | | | | — | 0.090[*] | −0.055 | −0.028 |
| 7. Regular strategy game play | Pearson's r | | | | | | — | −0.019 | 0.038 |
| 8. Regular online game play | Pearson's r | | | | | | | — | 0.020 |
| 9. Deductive reasoning ability | Pearson's r | | | | | | | | — |

**Notes.**
[*]Denotes p value below Bonferroni-Holm adjusted value of 0.0014.

**Table 3  Evidence from Bayesian hypothesis testing.**

| | Participants who did not express preference or play game type | | | Participants who did express preference or play game type | | | Average Enhancement Effect[a] | | Enhancement Effect in Top Tier Journals[b] | |
|---|---|---|---|---|---|---|---|---|---|---|
| | Count | Mean | SD | Count | Mean | SD | $BF_{01}$ | $BF_{10}$ | $BF_{01}$ | $BF_{10}$ |
| Action game preference | 1,530 | 0.02 | 1.00 | 317 | −0.12 | 0.97 | 16.38 | 0.06 | 57.01 | 0.02 |
| Strategy game preference | 1,442 | −0.02 | 1.02 | 405 | 0.08 | 0.93 | 0.69 | 1.45 | 2.06 | 0.48 |
| Online game preference | 975 | −0.05 | 1.04 | 872 | 0.05 | 0.95 | 0.36 | 2.82 | 1.07 | 0.94 |
| Regular action game play | 1,551 | 0.02 | 1.00 | 192 | −0.07 | 0.99 | 8.61 | 0.12 | 29.15 | 0.03 |
| Regular strategy game play | 1,462 | −0.02 | 1.02 | 276 | 0.08 | 0.90 | 0.84 | 1.20 | 2.45 | 0.41 |
| Regular online game play | 1,019 | −0.04 | 1.02 | 630 | 0.01 | 0.97 | 2.97 | 0.03 | 9.84 | 0.10 |

**Notes.**
$BF_{01}$ denotes evidence favoring the Null hypothesis. $BF_{10}$ denotes evidence favoring the alternative hypothesis.
[a]Average effect size for quasi-experiments on measures of reasoning and intelligence ($d = 0.24$; *Powers et al., 2013*).
[b]Average effect size for quasi-experiments on measures of general cognitive abilities published in top-tier journals ($d = 0.85$; *Powers et al., 2013*).

of 0.24, effect size for quasi-experiments on measures of reasoning and intelligence, and a second prior effect size for the enhancement hypothesis as reported in top-tier journals (Cohen's d of 0.85; *Powers et al., 2013*). Results provided very strong support for the null hypothesis over the alternative for action games ($BF_{10} = 0.06$; $M_0 = 0.02$, $SD_0 = 1.00$, $M_1 = -0.12$, $SD_1 = 0.97$) and equivocal support for alternative hypothesis for those who preferred strategy games ($BF_{10} = 1.45$; $M_0 = -0.02$, $SD_0 = 1.02$, $M_1 = 0.08$, $SD_1 = 0.93$), or online multiplayer games ($BF_{10} = 2.815$; $M_0 = -0.5$, $SD_0 = 1.04$, $M_1 = 0.05$, $SD_1 = 0.95$). In examining the robustness of these Bayes factors it is clear that evidence

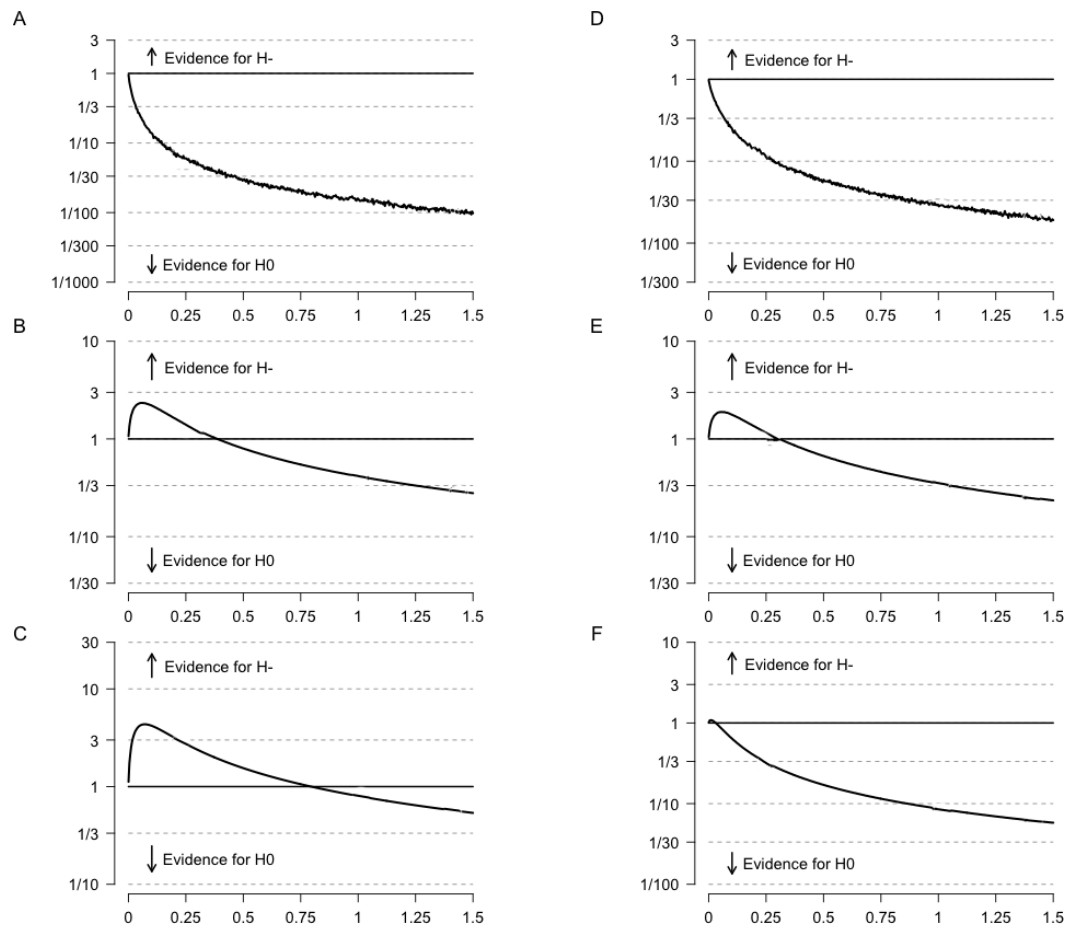

**Figure 1** **Bayes factors for the comparison of electronic gaming to reasoning ability as a function of the scale parameter of the Cauchy prior for effect sizes ranging from 0.0 to 1.5 under the alternative hypothesis.** Note. Equal variances are assumed. (A) through (C) represent game preference, and (D) through (F) represent regular game play. (A) Action games. (B) Strategy games. (C) Online games (D) Action games. (E) Strategy games. (F) Online Games.

for effects larger than the average in the literature are also not supported. Results derived using the larger effect sizes reported in top-tier journals for the enhancement effect, $d = 0.85$, appeared less likely as evidence was 1.25 and 57.01 times more likely to have been observed under the null hypothesis than under the gaming-enhancement hypothesis (see Table 3). Figure 1A–1C, present relative evidence for the gaming enhancement hypothesis for effect sizes ranging from a 0.0 to 1.5. Taken together with the focused hypothesis tests, these results indicated participants' preferences for these specific types of games were not reliably linked to their deductive reasoning ability in these data.

## Regular game play and reasoning ability

To examine the relations between regular action, strategy, or online game play and reasoning ability, three additional Bayesian hypothesis tests were conducted following the procedure used for game preferences. These models evaluated the relative evidence for the null hypothesis as well as the gaming-enhancement hypothesis that postulates

that playing these games for more than five hours each week would be associated with players' reasoning abilities. Results provided moderate support for the null over the alternative hypothesis for action games ($BF_{10} = 0.12$; $M_0 = 0.02$, $SD_0 = 1.00$, $M_1 = -0.07$, $SD_1 = 0.99$), equivocal evidence for strategy games ($BF_{10} = 1.120$; $M_0 = -0.023$, $SD_0 = 1.02$, $M_1 = 0.08$, $SD_1 = 0.90$), and equivocal to moderate evidence for the null over the alternative hypothesis for multiplayer online games ($BF_{10} = 0.34$; $M_0 = -0.4$, $SD_0 = 1.04$, $M_1 = 0.01$, $SD_1 = 0.97$). Robustness checks using effect size for the gaming enhancement hypothesis as reported in top-tier journals ($d = 0.85$; *Powers et al., 2013*) indicated evidence against the this hypothesis. Data were between 2.45 and 29.15 times more likely to be observed under the null hypothesis. Figure 1D–1F present relative evidence for the gaming enhancement hypothesis for effect sizes ranging from of 0.0 to 1.5. Taken together with the previous results, these findings suggest it is unlikely that regular active play of these games is systematically related to higher levels of general reasoning abilities.

## DISCUSSION

### Overview

The promise that electronic games might positively influence human cognition is one that generates intense public, corporate, and scientific interest. The present research drew on a large sample of school-aged children and considered both their electronic gaming and cognitive performance to test of the gaming-enhancement hypothesis. Of central interest was the nature of the potential relations between children's self-reported preferences and gaming habits, and performance on a widely used test of deductive reasoning ability. Contrary to our expectations, the results did not provide substantial evidence in support of the idea that the complex and interactive experiences provided by commercially available games generalize to functioning outside of gaming contexts. In most cases, the evidence was in favor of the null hypothesis over this account.

We hypothesized that those who express preferences for action, strategy, and multiplayer online games, modes of play would show modestly higher levels of cognitive performance as seen in previous smaller-scale studies (*Basak et al., 2011*). In contrast to what was expected, we found equivocal to very strong evidence favoring the null hypothesis over this prediction. Second, we hypothesized that regular engagement with games—playing five or more hours a week of action, strategy, and multiplayer online games—would be linked with better reasoning ability. Results from our analyses did not support this prediction. Evidence for regular strategy game players were equivocal, but ranged from moderately to strongly in favor of the null for these types of games. Taken together, our findings disconfirmed the gaming-enhancement hypothesis, especially in terms of the larger effects reported in top-tier journals.

Our approach carries a number of strengths that should inform future studies of gaming effects. First, evidence from this study relied on data provided by more than 1,800 young people, a sample over 50 times larger than the average for studies examining gaming and cognition (*Powers et al., 2013*). If research is to sort out the cognitive dynamics of play, larger and more robust sampling is needed. Second, the Bayesian hypothesis testing

approach we adopted used open source software (JASP; *JASP Team, 2016*) and allowed our study to quantify evidence for the both the null hypothesis as well as a plausible alternative based on the existing literature. Although the analysis plan for this study was not registered before the data were known, the framework provides valuable empirical data that researchers can use as the basis, or prior, to inform their own pre-registered designs. Finally, if indeed scholars are increasingly skeptical of corporate attempts to sell games based on their purported upsides (e.g., *Allaire et al., 2014*), this vigor should be extended to all scientific inquiry in this area, for example by making the materials, data, source and analysis code openly available. Future research making both positive and negative claims regarding the effects of gaming on young people should do likewise.

## Limitations and future directions

The present study presents a number of limitations that suggest promising avenues for future work. First, because the data under study were cross-sectional they capture an easy to interpret pattern of results that represent a snapshot in time. The data structure did so at the expense of being able to draw causal inferences, and a complementary approach would examine long-term salutary effects on cognition, ideally as a function of experimental manipulations of game exposure. Second, data regarding participants' game preferences and regular play were collected through self-report. Research indicates that some participants, and young people in particular, provide exaggerated data when it comes to taboo activities such as sexual habits and drug use (*Robinson-Cimpian, 2014*). It is possible that the average levels of engagement reported by our participants disguised interesting patterns of engagement which merit inquiry. For example, infrequent periods of high engagement (e.g., binge-playing) might have its own special relations with reasoning abilities. If so, an experience-sampling based approach would be needed to assess both between- and within-person variability with respect to the relation between gaming and reasoning ability. Finally, the present study only used a single assessment of general cognitive abilities, the Ravens Progressive Matrices task (*Raven, 2000*). There are many other facets to intelligence and executive control that might be more sensitive to influence by regular electronic gaming. Measures of naïve reasoning (*Masson, Bub & Lalonde, 2011*), short and long-term memory (*Melby-Lervåg & Hulme, 2013*), audio processing (*Liu & Holt, 2011*) might be more liable to be influenced by gaming. If the gaming-enhancement hypothesis is not broadly accurate, it may find empirical support under conditions where these alternative aspects of intelligence and reasoning abilities are under study.

## Closing remarks

The promise that popular games can enhance cognitive skills is an alluring one. Our findings suggest there is no relation between interest in, or regular play of, electronic games and general reasoning ability. As such, we advise that future research examining the potential influences of gaming contexts on players should pre-register their analysis plans or follow the registered reports process (e.g., *Chambers, 2013*; *Elson, Przybylski & Krämer, 2015*). Such steps would go a long way to reduce researcher degrees of freedom

which might, along with publication bias, affect conclusions drawn about the effects of gaming and cognitive enhancement (*Feynman, 1974*; *Gelman & Loken, 2013*; *Nissen et al., 2016*). While the research presented here might be further informed by additional work conducted to these standards, our findings above offer an early exploration of the gaming-enhancement hypothesis which is well-powered and guided by open-science.

### Funding

This research was partially funded by a John Fell Fund Grant (CZD08320) through the University of Oxford to Dr. Przbylski, and a joint grant awarded by the Ministry of Education, Singapore and the Media Development Authority f (EPI/06AK) to Professor Wang. The funders had no role in study design, data collection and analysis, decision to publish, or preparation of the manuscript.

### Grant Disclosures

The following grant information was disclosed by the authors:
John Fell Fund: CZD08320.
Ministry of Education, Singapore and the Media Development Authority: EPI/06AK.

### Competing Interests

The authors declare there are no competing interests.

### Author Contributions

- Andrew K. Przybylski conceived and designed the experiments, performed the experiments, analyzed the data, contributed reagents/materials/analysis tools, wrote the paper, prepared figures and/or tables, reviewed drafts of the paper.
- John C. Wang conceived and designed the experiments, performed the experiments, contributed reagents/materials/analysis tools, wrote the paper, reviewed drafts of the paper.

### Ethics

The following information was supplied relating to ethical approvals (i.e., approving body and any reference numbers):

Ethical clearance for the Effects of Digital Gaming on Children and Teenagers in Singapore project (EDGCTS) was sought and granted through the Institutional Review Board of Nanyang Technological University in Singapore. The Research and Ethics Committee at the Oxford Internet Institute conducted ethical review for secondary data analysis on the EDGCTS dataset (SSH/OII/C1A-16-063).

### Data Availability

All study materials and data are available for download using the Open Science Framework: https://osf.io/je786/.

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
