# Peer review of "A large scale test of the gaming-enhancement hypothesis"

_PeerJ, doi:10.7717/peerj.2710_

## Round 0.1 · original submission · Major Revisions

Dear Andrew,

Two reviewers with substantial expertise have examined your submission and provided critiques for consideration in the review process. The reviewers express interest in your submission on various accounts and offer encouragement for this line of inquiry. I agree with the positive sentiment expressed in the reviews. The investigation appears to have been well-conducted and manuscript was generally well-prepared for entry into the PeerJ review process.

Reservations are, nonetheless, evident in the reviewers' critiques. Their observations are presented with clarity so I'll not risk confusing matters by belaboring or reiterating their comments. While I might quibble with the occasional point, I note that I regard the reviewers' opinions as substantive and well-informed. I believe that all of the highlighted reservations require contemplation and appropriate attention in revising the document if it is to contribute appropriately to PeerJ and the extant literature.

I ask that all reviewer observations be addressed in revising the manuscript. How and where the reviewers’observations are addressed (or rebutted) should be explained on a point-by-point basis. Some matters will require little more than minor editing. Other matters are likely to require much more substantial contemplation and effort. I look forward to reading your revised submission and the accompanying line-by-line responses. I anticipate that both will make for interesting reading.

I note that I would reject your manuscript at this point if I was certain that the difficulties identified by the reviewers were impossible to overcome. Please understand, however, that this opportunity to revise and resubmit the manuscript does not guarantee that it will ultimately be accepted for publication. The revised manuscript and your line-by-line responses will be sent out for further reviewer input to ensure that my decision is appropriately informed.

Tsung-Min Hung, Ph.D.
PeerJ editor
Distinguished professor,
Department of Physical Education,
National Taiwan Normal University

·

Basic reporting

The article is clearly written and for the most part clearly described. I should caveat my review by declaring that I am not a specialist in this field, so I cannot comment with authority on the comprehensiveness of the literature review (although the work, as described, sounds appropriate to me).

Minor:
Line 194, "combing" > "combining" ?
Line 194, should "games" be inserted after "kinds of" ?
Lines 109-112: it would be helpful to provide more detail about how this meta-analytic effect size estimate of d=0.24 was arrived it, what kinds of tests it relates to, and how this relates to the average N of 32 in previous studies (as publication bias, combined with overall small N, might be expected to yield a higher meta-analytic estimate of the effect size).

Experimental design

I have a number of queries about the analysis approach.

1. Were there any multiple comparisons corrections applied to the frequentist correlational analyses reported in the Preliminary Analyses section? (Also, Line 211: the term "significantly" should be inserted in the sentence "Gender was not <significantly> related...")

2. I'm afraid I got confused with the "one-sided Bayesian independent samples t-tests" (Lines 213-240). I was expecting to see a series of Bayesian correlations (e.g. Wetzels and Wagenmakers http://dx.doi.org/10.3758/s13423-012-0295-x), corresponding to the frequentist correlation coefficients in table 2 for game preference vs RPM reasoning ability (3 vs 9; 4 vs 9; 5 vs 9) and regular game play (6 vs 9, 7 vs 9, 8 vs 9). Instead, the authors appear to have compared two independent groups of participants, but I don't understand what these groups could be, and the means are not reported. These sections should be clarified.

3. For completeness, I recommend providing full details of the range of priors used in the various robustness tests, with the results presented accordingly in a figure or table.

4. Line 218-219: "Results provided very strong support for the alternative hypothesis over the null for action games (BF10 = 0.061)" >> Shouldn't this be the other way around?

5. It would be helpful to provide more information about the procedure for standardising the RPM scores by age: (Line 175) "Age-adjusted reasoning scores were created for each participant by standardizing individual performance on the RPM by participant age in line with best practices"

Validity of the findings

The results are interpreted consistently with the evidence, and I especially applaud the authors' commitment to open research practices by making their data and analysis scripts publicly available.

The authors note the importance of pre-registration. As a side point, I am curious to know whether the secondary analyses reported herein were pre-registered?

·

Basic reporting

Intro-
1. One of the strengths of current study is the examination for the relations between specific types of game (i.e., action, multiplayer online, strategy) and reasoning ability. Therefore, efforts should be exercised on describing extant knowledge regarding this specific relation and why examination for this relation so important to the knowledge base when drafting the manuscript. Nevertheless, as far as I am concern, these issues have not been clearly elaborated by the authors.
2. Another noteworthy issue of current study should be that you target on school-aged children, population which is rarely focused by related studies. Thus, efforts should be exercised on noting why examination for this particular population important to the knowledge when drafting the manuscript. It might be helpful to address this issue with developmental perspectives.
3. The authors assume that individuals who play challenging games for more than five hours per week would demonstrate superior reasoning ability, which implies that you assumed regular engagement to games plays a role in mediating the relationship between gaming experience and cognitive ability. Accordingly, I would like to see the authors elaborate how regular/irregular engagement might modulate the relation.
4. The flow and logic of paragraph 3 looks somewhat strange to me. The authors use a topic sentence indicating that the predisposition for specific kinds of games may relate to cognitive performance; however, personally I would like to see the authors provide more details to justify this argument. Further, at the end of the paragraph, the statement “Unfortunately, many of ……the gaming enhancement hypothesis” is somewhat illogical. I believe this statement could be raised somewhere else.
5. I looks that the Bayesian approach is one of your strength compared with previous works. Thus, I would like to see the authors provide more information on how this approach of analysis contributes to your study.

Results-
In table 2, please notify/mark the significant pairs.

Discussion-
It is hard to see the contribution of current study if the authors only reported that the findings failed to support the hypothesis. I believe more discussion should be exercised regarding the findings. Currently the Discussion section is very weak.

Minor issues-
1. There are several typing errors throughout the manuscript. Please correct them.
2. Though I am not a native English speaker, I believe some wordings should be corrected in the manuscript.

Experimental design

There are several issues warrant further elaborations in Method-
1. No description for statistics?
2. What exactly the age range of your participants is?
3. Does the assessment for reasoning ability suitable to all of your participants?
4. Why use the RPM task as assessment tool for reasoning ability?
5. Any reports for the validity and reliability of your questionnaire/assessment?

Validity of the findings

I believe the validity of the findings can be strengthened if the authors clarify issues I raised in Intro and Method.

Additional comments

The current study examined the relation between gaming experience and cognitive ability (i.e., reasoning). The main findings were that there was no significant relation between gaming experience and reasoning ability. The current manuscript, indeed, targets on an interesting and relevant topic; however, this manuscript is not well-written. The Intro does not provide sufficient information regarding extant findings as well as the rationale for current examination. Second, more details should be provided in the Method. Third, the Discussion is very weak with current format, and considerable efforts should be made to refine it. Overall, despite the current manuscript addresses a relevant topic and is consistent with the scope of PeerJ, it is suffered from several constraints in the way data was reported/elaborated. The authors should try their best taking care of the issues I raised to refine their work.

---

## Round 0.2 · Minor Revisions

I have now received two re-reviewers’ comment and both reviewers were generally satisfied with your reply and revisions from previous comments. However, one reviewer has pointed out 2 minor issues that require your additional attention. Please address these issues and provide a point by point reply in addition to the revised manuscript.

Tsung-Min Hung, Ph.D.
PeerJ editor
Distinguished professor
Department of Physical Education
National Taiwan Normal University

·

Basic reporting

N/A

Experimental design

N/A

Validity of the findings

N/A

Additional comments

The authors have responded thoroughly to my comments.

·

Basic reporting

Minor issues-
1. Shouldn't "longer-term memory" be revised as "long-term memory"?
2. Please provide evidence that supports the statement "Measures of fluid reasoning, short and longer-term memory, and verbal intelligence might be more liable to be influenced by gaming"

Experimental design

no comments

Validity of the findings

no comments

Additional comments

Dear authors,
I am satisfied with most of the revisions. Only two minor concerns are further addressed. Good jobs!

---

## Round 0.3 · Minor Revisions

I have read through your reply to the reviewer's comment and your revised manuscript. I am satisfied with your response. However, I spoted 2 minor wording issues that require your attention.

Here they are.

P8, L215, The sentence of “Because gender was not associated with our target outcome, which was centered on age, neither age or gender were considered as covariates in hypothesis testing” read weir. Do you mean …..neither age “nor” gender….. ?

P12, L266-267, please delete the word of “the” from the sentence “In contrast to what was expected, we found equivocal to very strong evidence favoring the null hypothesis over the this prediction”. I believe that is a typo.

I'll accept your revised manuscript after you address these issues and resubmit a corrected manuscript. Thank you for your patience and look forward to receiving your finalized manuscript soon.

Tsung-Min Hung, Ph.D.
PeerJ editor
Distinguished professor
Department of Physical Education
National Taiwan Normal University

---

## Round 0.4 · accepted · Accept

I have read through your reply to my comment and your revised manuscript. I am satisfied with your response and decided that your manuscript is suitable to be published in PeerJ. You and your coauthors have my congratulations. Thank you for choosing PeerJ as a venue for publishing your research work and I look forward to receiving more of your work in the future.

Tsung-Min Hung